# Notch, RORC and IL-23 signals cooperate to promote multi-lineage human innate lymphoid cell differentiation

Carys A. Croft [1], Anna Thaller [1], Solenne Marie[1], Jean-Marc Doisne [1], Laura Surace[1], Rui Yang[2], Anne Puel[3,4,5], Jacinta Bustamante[3,4,5], Jean-Laurent Casanova[2,3,4,5,6] & James P. Di Santo [1] ✉

Innate lymphoid cells (ILCs) include cytotoxic natural killer cells and distinct groups of cytokine-producing innate helper cells which participate in immune defense and promote tissue homeostasis. Circulating human ILC precursors (ILCP) able to generate all canonical ILC subsets via multi-potent or uni-potent intermediates according to our previous work. Here we show potential cooperative roles for the Notch and IL-23 signaling pathways for human ILC differentiation from blood ILCP using single cell cloning analyses and validate these findings in patient samples with rare genetic deficiencies in *IL12RB1* and *RORC*. Mechanistically, Notch signaling promotes upregulation of the transcription factor *RORC*, enabling acquisition of Group 1 (IFN-γ) and Group 3 (IL-17A, IL-22) effector functions in multi-potent and uni-potent ILCP. Interfering with RORC or signaling through its target IL-23R compromises ILC3 effector functions but also generally suppresses ILC production from multi-potent ILCP. Our results identify a Notch->RORC- > IL-23R pathway which operates during human ILC differentiation. These observations may help guide protocols to expand functional ILC subsets in vitro with an aim towards novel ILC therapies for human disease.

Innate lymphoid cells (ILC) are a family of lymphoid effector cells with notable similarities to T cells but lacking rearranged antigen receptors. ILCs can rapidly produce soluble factors (cytokines, chemokines) in response to environmental stimuli and actively participate in steady-state immune homeostasis as well as in the early immune response to infection, inflammation and tissue stress[1–3]. Based on expression of transcription factors and cytokine production, ILCs in both mice and man have been categorized into five functional groups that mirror known helper and cytotoxic T cell subsets[4,5]. NK cells, which express Eomesodermin (EOMES) and T-box transcription factor 21 (TBX21 encoding T-BET), are highly cytotoxic effector cells that also produce interferon gamma (IFN-γ) and tumour necrosis factor (TNF) and

thereby represent innate counterparts of CD8⁺ cytotoxic T-cells[6]. Related ILC1 are a non-cytotoxic subset of TNF and IFN-γ producing innate cells that also depend on T-BET (but not EOMES) and may represent innate homologues of Th1cells[7,8]. Group 2 ILCs require GATA-binding protein 3 (GATA-3) for development and are able to produce type 2 cytokines (interleukin (IL)−5, IL-9, IL-13) as well as the EGF-related factor amphiregulin and can be considered as innate homologues of Th2 cells[9–12]. Group 3 ILCs are ROR related orphan receptor gamma t (RORγT)-dependent cells which produce IL-17A and IL-22 and are functionally analogous to Th17/22 cells[13,14]. Lastly, lymphoid tissue inducer cells (LTi) are a distinct fetal ILC3 subset that promote secondary lymphoid tissue development in the embryo[15,16].

[1]Institut Pasteur, Université Paris Cité, Inserm U1223, Innate Immunity Unit, Paris, France. [2]St. Giles Laboratory of Human Genetics of Infectious Diseases, Rockefeller Branch, Rockefeller University, New York, NY, USA. [3]Laboratory of Human Genetics of Infectious Diseases, Necker Branch, INSERM, UMR 1163 Paris, France. [4]Imagine Institute, Université Paris Cité, Paris, France. [5]Study Center for Primary Immunodeficiencies, Necker Hospital for Sick Children, AP-HP, Paris, France. [6]Howard Hughes Medical Institute, New York, NY, USA. ✉e-mail: james.di-santo@pasteur.fr

The signals that drive primary cell fate decisions from mouse or human ILCP remain poorly defined[17,18]. Earlier studies characterized CD34+ hematopoietic cells from human secondary lymphoid tissues and defined distinct NK/ILC-lineage precursors that could give rise to mature ILCs in vitro[19–21]. We identified a circulating human ILC precursor (ILCP) with a Lineage−CD34−CD7+CD117+CRTh2− phenotype which could generate all ILC subsets in vitro (using single cell cloning assays and bulk culture), as well as in vivo in a humanized mouse model[22]. At steady state, human ILCP lack expression of ILC group-defining transcription factors and do not produce cytokines, even following activation[22]. Nevertheless, human ILCP are present in numerous lymphoid and non-lymphoid tissues, existing in an epigenetically "poised" state ready for further expansion and differentiation to mature functional ILCs[22]. Based on these observations, we proposed an 'ILC-poiesis' model whereby circulating ILC precursors might enter tissues, and undergo a process of activation, proliferation and differentiation dictated by signals present in the tissue environment[22,23]. While ILC-poiesis can help explain some characteristics of distinct ILC 'repertoires' found under steady-state conditions and after infection[24], the roles for identified signaling pathways (Notch, RORC, cytokines) that promote Group 1, Group 2 or Group 3 ILC differentiation from blood ILCP remain to be fully characterized.

The transcription factor RORγT and the cytokine IL-23 are important drivers of 'type 3' T cell differentiation (involving Th17 and Th22 cells)[25,26]. Th17 cells are absent in *RORC*-deficient mice, and interestingly ILC3 generation is also ablated, highlighting the developmental similarities of mouse ILC3 and Th17 cells[13,15]. Similarly, patients with genetic deficiencies in *RORC*, or in IL-23 signaling via loss of function (LOF) mutations in *IL12RB1* or *TYK2*, have defects in Th17 cell generation and function, and consequently are susceptible to infections with fungal pathogens[26,27]. Additionally, IL-23 signaling in humans promotes type 1 immunity, as *IL23R−/−* patients have reduced IFN-γ expression, clinically manifesting as Mendelian susceptibility to mycobacterial disease (MSMD)[28–30]. Concerning human ILC generation, mutations in *RORC* have been previously reported to impair ILC3 development in human and have severe effects on the development of peripheral lymph nodes[22,27], while mutations in the IL-12/IL-23 signaling axis impair the ability of ILC2 to acquire IFN-γ expression upon IL-12 signaling[31].

The precise contributions of these factors to ILC development have not been clearly defined. Here we identify an essential role for *RORC* in the generation of human ILC3, and an important contribution of downstream IL-23 signaling in the acquisition of type 1 and type 3 effector functions during human ILC differentiation from circulating ILCP. Our results further refine our understanding of the regulation of human ILC differentiation and the roles Notch and IL-23 signaling play in this process.

## Results

### Notch signaling and IL-23 promote ILC differentiation from human ILC precursors

We previously showed that peripheral blood human ILCP can give rise to diverse mature ILC subsets in vitro following bulk culture or single cell cloning on OP9 stromal cells in the presence of cytokines[22]. ILCP were identified in the peripheral blood as Lin−CD7+CD94−NKG2A−CD16−CD127+CRTH2−CD117+ (Supplementary Fig. 1a) and while the majority were uniformly CD25+, CD45RA+ and CD200R1+ (in line with previous reports[22,32,33]), human ILCP were heterogeneous for CD56, CD62L and CD161 expression (Supplementary Fig. 1b). To better understand the individual roles for stromal-derived signals (Notch) and cytokines (IL-23) in promoting ILC differentiation, we analyzed progeny from short-term bulk cultures of human ILCP. Blood ILCP were cultured in wells pre-seeded with either OP9 or OP9-DLL4 stroma in media containing IL-2, IL-7 and IL-1β with or without IL-23. After 5–7 days, expanded cells were electronically gated to exclude

NK cells and remaining cells were tested for transcription factors associated with development of ILC1 (T-BET), ILC2 (GATA-3) and ILC3 (RORC) as well as for their ability to produce cytokines upon short-term pharmacological stimulation with phorbol 12-myristate acetate/ionomycin (P/I).

Concerning ILC1 differentiation, IL-23 boosted IFN-γ production in cells cultured on OP9 or OP9-DLL4 stroma, with the highest percentage of IFN-γ-producing cells under IL-23/DLL4 conditions (Fig. 1a, Supplementary Fig. 2a, Supplementary Table 1). Interestingly, this was not associated with significant increases in T-BET (Fig. 1b, c, Supplementary Fig. 2b, Supplementary Table 1) and is consistent with the ability of IL-23 to promote IFN-γ production in humans[29,34]. For ILC2 differentiation, IL-13 production varied greatly between healthy donors but was not significantly modulated by exposure to either DLL4, while a small but significant decrease was observed with the addition of IL-23 only in the absence of Notch signaling (Fig. 1d, Supplementary Fig. 2c, Supplementary Table 1). In contrast, GATA-3 levels were significantly higher in cultures grown on OP9-DLL4 (Fig. 1e, f, Supplementary Fig. 2d, Supplementary Tables 1–2). Concerning ILC3-associated cytokines, IL-17A was produced at low levels in most donors and was only appreciably expressed in OP9-DLL4 cultures in the presence of IL-23 (Fig. 1g, Supplementary Fig. 2e, Supplementary Table 1). In contrast, IL-22 production was significantly increased in bulks cultured on OP9-DLL4 and was highest in bulk cultures on OP9-DLL4 in the presence of IL-23 (Fig. 1h, Supplementary Fig. 2f, Supplementary Table 1). ILC3-related RORγT expression was also significantly higher in ILCP cultured on OP9-DLL4 (Fig. 1i, Fig. Supplementary Fig. 2g, Supplementary Table 1), with an average 3-fold increase in the percentage of RORγT+ cells between ILCP from the same donor cultured on OP9-DLL4 as compared to OP9 cells (Fig. 1j, Supplementary Table 2). Taken together, these results suggest an effect of Notch signaling via DLL4 on RORC/RORγT expression in developing ILC and a synergistic role for IL-23 and Notch signaling for the generation of Group ILC1 and ILC3 from peripheral blood ILCP.

### Human ILCP differentiation in RORC-deficient patients

As DLL4 signaling could increase frequencies of RORγT+ cells generated from ILCP in bulk culture irrespective of cytokine combinations (Fig. 1i, j), we hypothesized that Notch-induced RORC expression might facilitate generation of certain ILC subsets (especially ILC3) from ILCP. We analyzed two MSMD patients with genetic deficiencies in the transcription factor RORC (encoding RORγT) (Supplementary Table 3)[26,27]. We first characterized circulating ILC populations in *RORC−/−* patients and were able to identify ILCP, ILC2 and NK cells (Fig. 2a, Supplementary Fig. 3a). While the frequency of NK cell subsets and total CD127+ILCs were within the normal range (Fig. 2b, Supplementary Fig. 3b), both patients had reduced ILCP frequencies (Fig. 2c, f), with increased proportions of blood ILC2 (Fig. 2d, g), resulting in an altered ratio of ILCP/ILC2 compared to healthy controls (Fig. 2e).

We examined the cell surface phenotype of ILCP in *RORC−/−* patients using CD56, CD161, CD25, CD200R, CD45RA and CD62L. While ILCP from *RORC−/−* donors did cluster with healthy donor ILCP following uniform manifold approximation and projection (UMAP) analyses (Fig. 2h–i)[35], they could be differentiated based on several markers including lower levels of CD117 and higher levels of CD56 (Supplementary Fig. 3c–d). In contrast, CD161, CD45RA, CD25, CD62L and CD200R1 expression on ILCP from *RORC−/−* patients paralleled observations made on control ILCP in previous reports[22,32,33] (Fig. 2h–i).

As ILC differentiation from *RORC−/−* patients had been previously assessed solely in bulk cultures, we next sought to clarify the role of *RORC* in the determination of primary cell fate. We therefore employed a previously described in vitro cell cloning system[22] based on OP9 stromal cells with exogenous cytokines, including IL-1β, IL-2, IL-7 and IL-23, to examine the developmental potential of single ILCs

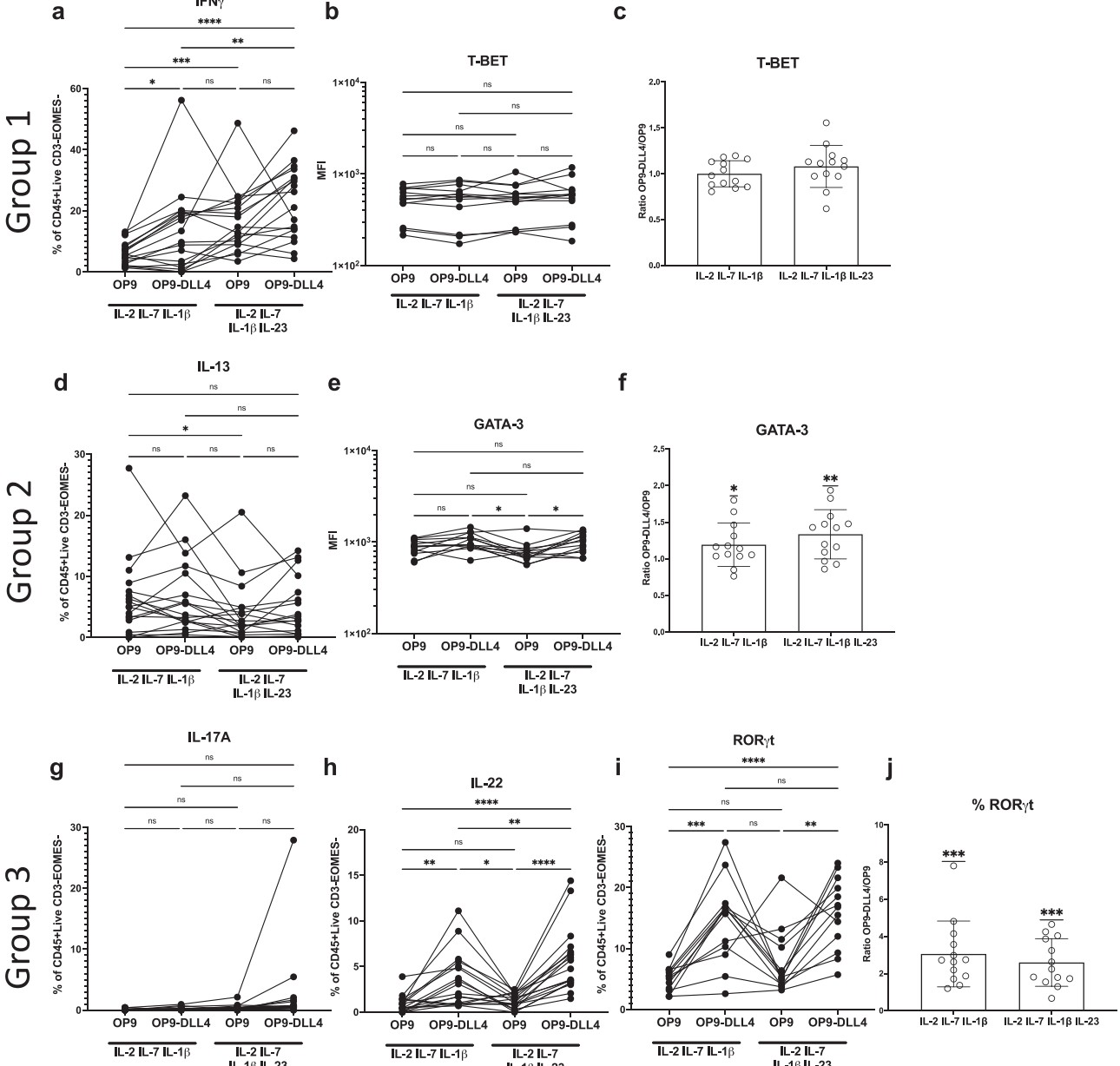

**Fig. 1 | Effects of Notch and IL-23 signaling for human ILCP differentiation.** ILC Group 1, 2 and 3 cytokine expression ($n = 18$; after P/I stimulation) and transcription factors ($n = 13$) were assessed in bulk ILCP cultures (Supplementary Fig. 2). Expression of Group 1 ILC associated factors (**a**) IFNγ (**b**) T-BET (**c**) and relative quantity of T-BET on OP9-DLL4 vs OP9. Expression of Group 2 ILC associated factors (**d**) IL-13 (**e**) GATA-3 (**f**) and relative quantity of GATA-3 on OP9-DLL4 vs OP9. Expression of Group 3 ILC associated factors (**g**) IL-17A (**h**) IL-22 (**i**) RORγt (**j**) and relative quantity of RORγt on OP9-DLL4 vs OP9. Data compared using (**a–b**, **d–e**, **g–i**) one-way repeated measures ANOVA using Tukey's multiple comparison's test, details in Supplementary Table 1 or (**c**, **f**, **j**) one sample T and Wilcoxon test, details in Supplementary Table 2. ns = not significant, *$p \leq 0.05$, **$p \leq 0.01$, ***$p \leq 0.001$, ****$p < 0.001$. Data are presented as (**a–b**, **d–e**, **g–i**) individual unique donors matched across conditions or (**c**, **f**, **j**) mean with SD of ratios of individual donors. Results from from 6 (**a**, **d**, **g–h**) or 4 (**b–c**, **e–f**, **i–j**) independent experiments. Source Data are provided as a Source Data file.

(Supplementary Fig. 4a). ILC clones were then classified based on cytokine expression. Group 1 ILCs, comprising both NK cells (EOMES+) or ILC1 (EOMES−) produced IFN-γ, ILC2 clones as IL-13-producers and ILC3 as IL-17A- and/or IL-22-producing clones (Supplementary Fig. 4b–f). As previously reported[22], some ILCP 'clones' harbored independent subsets of cells producing cytokines associated with different ILC groups; these 'multi-potent' clones were further designated as either IFN-γ and IL-13 producing Group 1/ILC2, IFN-γ and either IL-17A or IL-22 producing Group 1/ILC3, IL-13 and either IL-17A or IL-22 producing ILC2/ILC3, or IFN-γ, IL-13 and either IL-17A or IL-22 producing Group 1/ILC2/ILC3 (Supplementary Fig. 4h–k). As previously reported by us and others[22,32], a variable fraction of ILC clones did not produce any cytokines after stimulation and were considered as 'non-producers' (Supplementary Fig. 4g).

We analyzed ILCP-derived ILC clones from the above-described *RORC*−/− patients. Although cloning efficiencies from *RORC*−/− ILCP were severely reduced compared to healthy donors (Fig. 2j), the few *RORC*−/− clones that were generated on OP9-DLL4 stroma with IL-23 expressed exclusively IFN-γ or IL-13; none produced IL-17A or IL-22, in contrast to clones from healthy donors (Fig. 2k), consistent with our previous report[22]. These results confirm a RORC-independent pathway of human ILC1, ILC2 and NK cell development and an essential role for RORC in the differentiation of human ILCs that produce ILC3-signature cytokines.

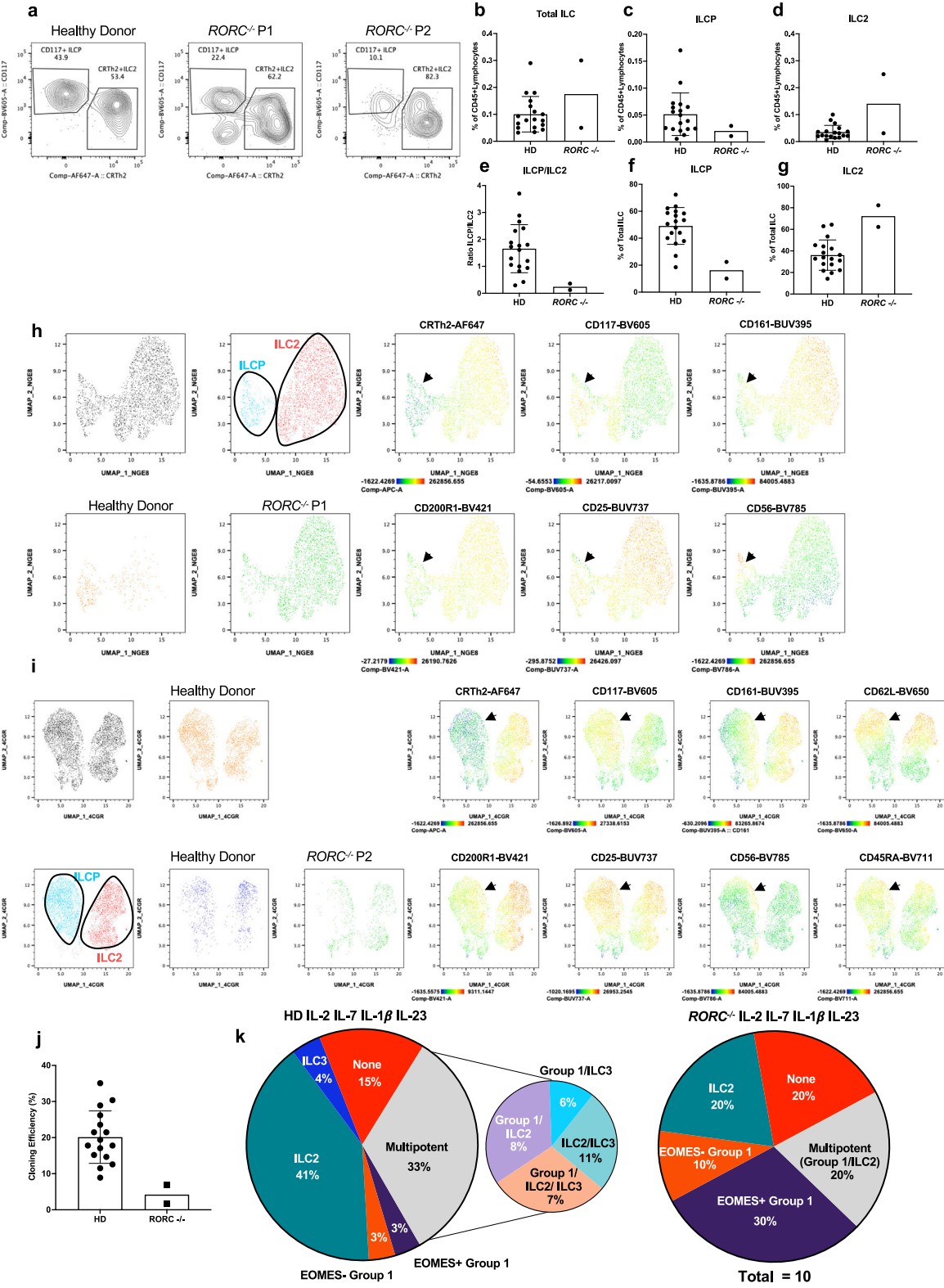

**Fig. 2 | Human ILC differentiation in *RORC*⁻/⁻ patients. a** Flow cytometric analysis of ILCs from a healthy donor and two *RORC*⁻/⁻ patients (gating strategy in Supplementary Fig. 1). Relative frequencies of **b** total ILC (**c**) ILCP (**d**) and ILC2 in CD45 + live lymphocytes. **e** Ratio of ILCP to ILC2. Relative frequencies of **f** ILCP and **g** ILC2 in total ILCs. **h–i** Uniform manifold projection (UMAP) analyses on healthy donors and *RORC*⁻/⁻ patients. **h** UMAP of experiment with P1 and healthy donor and (**i**) UMAP of experiment with P2 and healthy donors. Representative plots of staining can be found in Supplementary Fig. 3. **j** Cloning efficiency of healthy donors and *RORC*⁻/⁻ patients. **k** Clone distributions of *RORC*⁻/⁻ patients and healthy donors analyzed concurrently. **b–g, j** Data are represented as mean with SD, with each point corresponding to an individual unique donor. Results compiled from healthy donors (*n* = 18) and *RORC*⁻/⁻ patients (*n* = 2). Source Data are provided as a Source Data file.

## Pharmacological RORC inhibition alters ILC differentiation in bulk and clonal ILCP cultures

As access to ILCP from RORC-deficient patients was limited, we used an orthogonal approach to better understand how RORγT conditions ILC fate. We used the RORC inhibitor SR2211, which has been shown to modulate IL-23R and IL-17A expression in differentiating T cells[36]. We included this inhibitor in bulk ILCP cultures on OP9-DLL4 with IL-1β, IL-2, IL-7 and IL-23 and assessed its effects on transcription factor expression as well as P/I-induced cytokine expression. Addition of SR2211 did not significantly affect T-BET or GATA-3 expression, (Fig. 3b, d, Supplementary Fig. 5b, d, Supplementary Table 4) or the production of IL-13 from bulk cultured ILCP (Fig. 3c, Supplementary Fig. 5c, Supplementary Table 4). In contrast, the frequencies of mature ILCs producing IFN-γ or IL-22, or expressing RORγT were significantly reduced by SR2211 (Fig. 3a, f, g, Supplementary Fig. 5a, f–h, Supplementary Table 4). These results suggest that RORC inhibition modifies the generation of Group 1 and Group 3 ILCs in bulk cultured human ILCP.

We next assessed the impact of SR2211 on clonal ILCP differentiation. RORC inhibition increased (average of 15%) the frequency of uni-potent ILC2 clones, while it decreased (average of 20%) the generation of multi-potent ILC clones (Fig. 4a, Supplementary Table 5). Concerning the diversity of multi-potent clones generated, exposure

to SR2211 significantly decreased the frequency of ILC2/ILC3 clones and almost completely ablated the generation of ILC1/ILC3 clones, with only 1 donor able to generate any ILC1/ILC3 clones when cloned with SR2211, while a majority of donor ILCP (12 of 16) generated ILC1/ILC3 clones in control DMSO clonal cultures (Fig. 4b, Supplementary Table 5). While differentiation of uni-potent ILC1 was not obviously impacted by SR2211, and unipotent ILC3 were reduced only modestly (Fig. 4a, Supplementary Table 5), the overall frequency of IL-17A and IL-22 producing ILC clones was significantly reduced (Fig. 4d, Supplementary Table 5), reflecting the strong decrease at the level of multi-potent progenitors. The cumulative distributions of ILC clones generated from all donors were significantly different in the presence or absence of SR2211 (Fig. 4c, Supplementary Table 6). In line with results $RORC^{-/-}$ patients (Fig. 2j), SR2211 significantly reduced ILCP cloning efficiency in healthy donors (Fig. 4e, Supplementary Table 4). Collectively, these results identify multiple roles for RORγT in the generation of mature ILC3 from Notch plus IL-23-stimulated multi-potent ILCP.

## Effects of Notch signaling and IL-23 on clonal ILC differentiation from blood ILCP

As RORC promotes Th17 T cell differentiation through up-regulation of IL-23R[36–38], we hypothesized that IL-23 signals might represent the

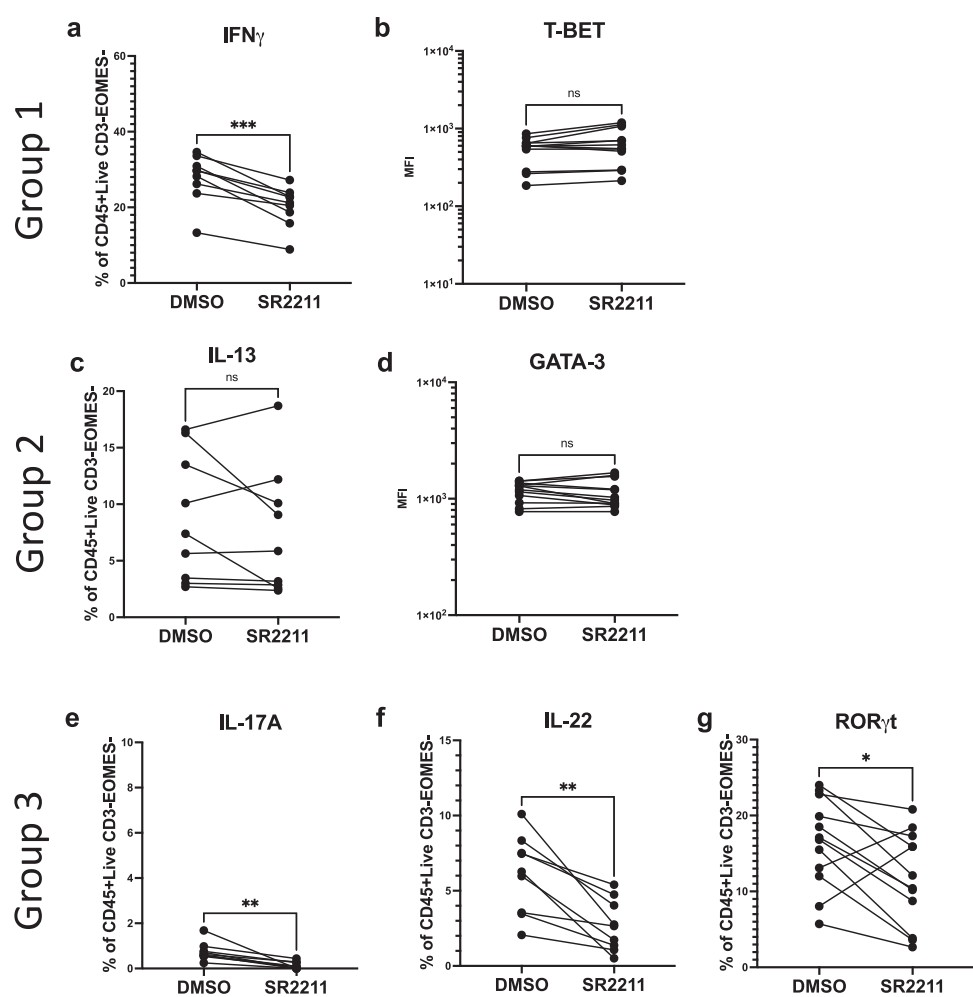

**Fig. 3 | Impact of RORC inhibition for ILCP differentiation.** Analysis of cytokine (n = 9) and transcription factor (n = 12) expression in differentiated human ILCP after addition of either 0.01% DMSO or SR2211 (10 uM) was assessed as described in Fig. 1. Representative plots in Supplementary Fig. 4. Expression of Group 1 ILC associated factors (**a**) IFNγ and **b** T-BET. Expression of Group 2 ILC associated factors (**c**) IL-13 (**d**) and GATA-3. Expression of Group 3 ILC associated factors (**e**) IL-17A and **f** IL-22 (**g**) RORγt. Data compared using paired T-tests (two-tailed), details in Supplementary Table 4. ns = not significant, *p ≤ 0.05, **p ≤ 0.01, ***p ≤ 0.001, ****p < 0.001. Data are presented as individual unique donors matched across conditions. Results from 3 (**a, c, e, f**) or 4 (**b, d, g**) independent experiments. Source Data are provided as a Source Data file.

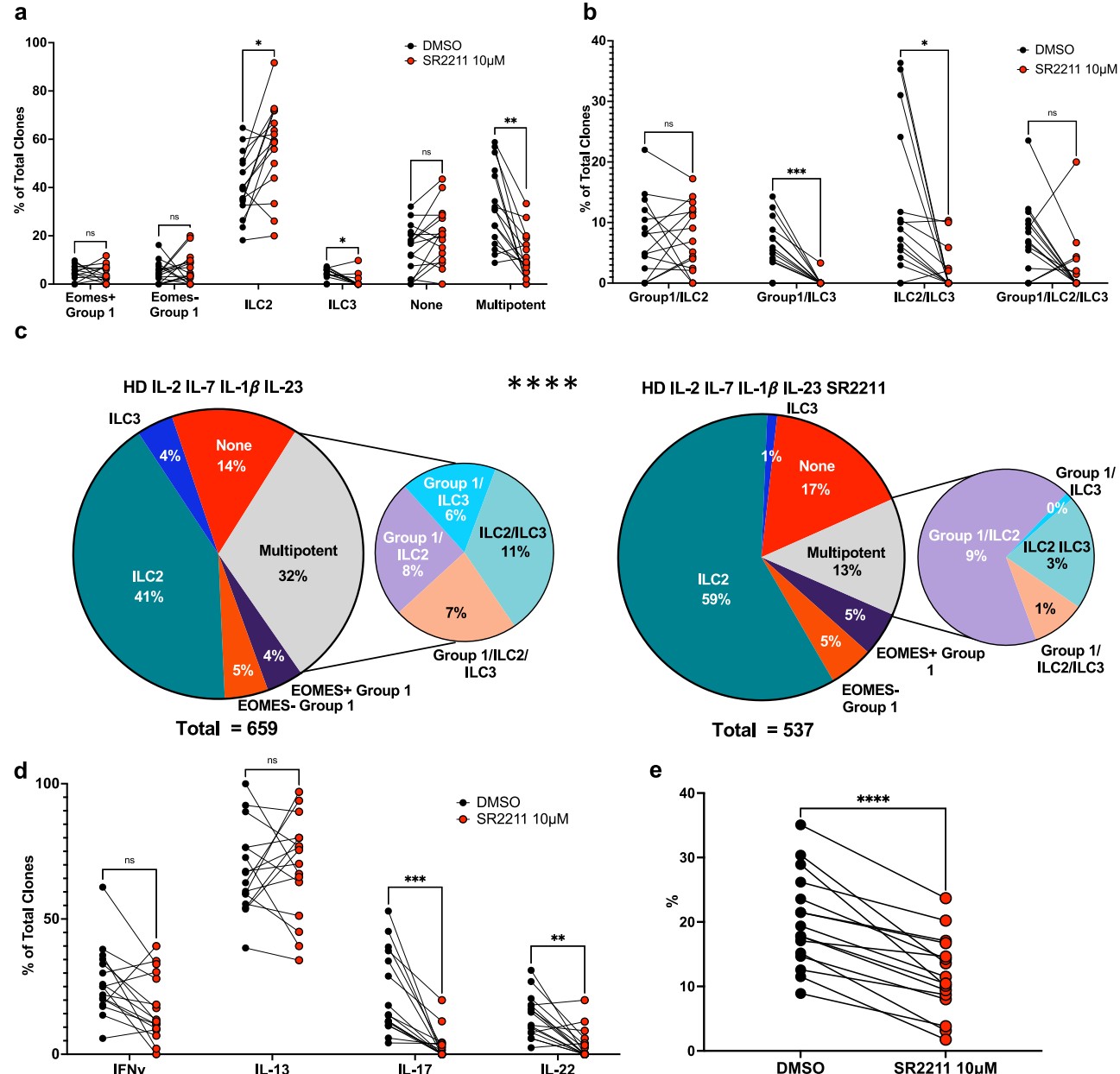

**Fig. 4 | Impact of SR2211 on clonal ILCP differentiation.** Single ILCP from healthy donors ($n = 16$) were cultured with either 0.01% DMSO (black circles) or SR2211 (10 $\mu$M) (red). **a** Frequencies of unipotent (EOMES + ILC1, EOMES-ILC1, ILC2, ILC3, None) and total multipotent clones per individual healthy donor. **b** Frequencies of types of multipotent clones. **c** Overall distributions of clones grown in either DMSO alone or SR2211. **d** Overall frequencies of all clones expressing IFNγ, IL-13, IL-17A and IL-22. **e** Cloning efficiency of donors grown in either DMSO or SR2211. Comparisons performed using (**a**–**b**, **d**) Two-way ANOVA with matching using Šidák's multiple comparisons tests, (details in Supplementary Table 5) (**c**) chi-square test (observed vs expected; details in Supplementary Table 6) and **e** paired t-test (two-tailed) (details in Supplementary Table 4). ns = not significant, *$p \le 0.05$, **$p \le 0.01$, ***$p \le 0.001$, ****$p < 0.001$. Data represented as individual donors matched across conditions, compiled from 7 experiments. Source Data are provided as a Source Data file.

primary mechanism by which Notch-induced RORC exerts its effects in human ILCP differentiation. To test this hypothesis, we cloned healthy donor ILCP on OP9 ($n = 20$) or OP9-DLL4 ($n = 19$) stroma with or without IL-23. By analyzing the frequencies of different ILCs arising from single ILCP in multiple healthy donors, we could make some generalizations about the independent roles for IL-23 and Notch signaling in human ILCP differentiation.

When culturing blood ILCP on OP9 stroma, we found that ILC2 was the dominant ILC subset generated irrespective of the presence or absence of IL-23 or Notch signaling (Fig. 5a, Supplementary Table 5). However, the overall frequency of ILC2 arising from ILCP was clearly reduced (average of 16%) when ILCP were cloned on OP9 stroma in the

presence of IL-23, and the frequency of multi-potent clones increased (average of 6%) in the presence of IL-23 (Fig. 5a, Supplementary Table 5). The types of multi-potent clones most influenced by IL-23 were those producing ILC3-related cytokines IL-17A or IL-22 (Fig. 5b, Supplementary Table 5), and the overall frequency of clones producing IL-17A was substantially increased (Supplementary Fig. 6a, Supplementary Table 5). The cumulative difference in the distributions of ILC clones generated on OP9 stroma with or without IL-23 was statistically significant (Fig. 5c, Supplementary Table 6; chi-squared = $p < 0.0001$). Importantly, IL-23 had no effect on cloning efficiency (total # clones obtained/total # of ILCP wells seeded; Supplementary Fig. 6i, Supplementary Table 4).

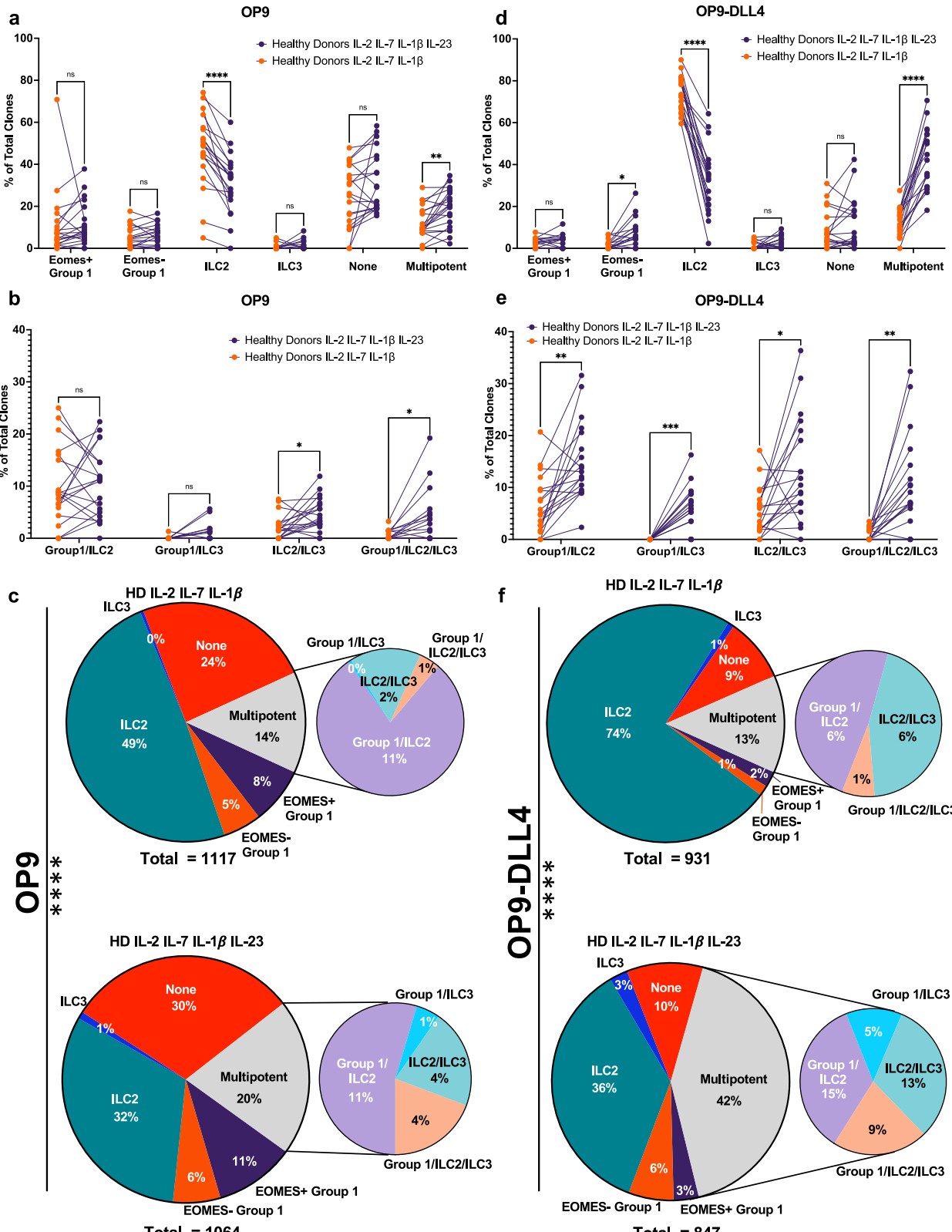

**Fig. 5 | Effects of Notch signaling and IL-23 on clonal ILC differentiation from blood ILCP.** Human ILCP cloning on OP9 (*n* = 20) or OP9-DLL4 (*n* = 19) stroma with (purple circles) or without (orange circles) IL-23 supplementation. **a** Frequencies of unipotent (EOMES + ILC1, EOMES-ILC1, ILC2, ILC3, None) and total multipotent clones per individual healthy donor, **b** Frequencies of types of multipotent clones and **c** Overall distributions of clones grown in the presence or absence of IL-23 on OP9 stroma. **d** Frequencies of unipotent and total multipotent clones per individual healthy donor, **e** Frequencies of types of multipotent clones and **f** Overall

distributions of clones grown in the presence or absence of IL-23 on OP9-DLL4 stroma. Data represented as individual matched values; each point corresponds to an individual donor. Comparisons performed using (**a**–**d**) Two-way ANOVA with matching using Šidák's multiple comparisons test, details in Supplementary Table 5 and **e**, **f** chi-square test (observed vs expected; details in Supplementary Table 6). ns = not significant, *$p \leq 0.05$, **$p \leq 0.01$, ***$p \leq 0.001$, ****$p < 0.001$. Data represented as individual donors matched across conditions. Source Data are provided as a Source Data file.

These results suggest that IL-23 promotes the growth of multi-potent ILC clones producing ILC3 cytokines at the expense of unipotent ILC2 clones. Still, the frequencies of IL-17A and IL-22 ILC clones on OP9 were quite low (Supplementary Fig. 6a, Supplementary Table 5). As ILCP bulk culture on OP9-DLL4 increased both RORγT expression and IL-22 production (Fig. 1g–j, Supplementary Table 1), we reasoned that enhanced Notch signaling might facilitate Group 3 features during ILCP differentiation, especially in the presence of IL-23. ILCP cloned on OP9-DLL4 stroma with IL-23 showed increased frequencies (average of 23%) of multi-potent clones producing ILC3-related cytokines and showed significantly decreased frequencies of ILC2 clones (average of 41%; Fig. 5d–f, Supplementary Table 6). Thus, the effect of IL-23 to promote multi-potent ILC clones and reduce ILC2 clones (as seen on OP9 stroma) was retained in the presence of DLL4 stimulation. Moreover, cloning of DLL4-stimulated ILCP in the presence of IL-23 showed an increased (>3-fold) frequency of total IL-17A and IL-22 producing clones (Supplementary Fig. 6b, Supplementary Table 5). Interestingly, OP9-DLL4 cultures with IL-23 generated a modest (average of 5%) but significant increase in EOMES⁻Group 1 clones (Fig. 5d, Supplementary Table 5). Finally, Notch signaling with IL-23 resulted in a significant reduction in ILC 'non-producer' clones (Supplementary Fig. 6d, Supplementary Table 5). These results suggest that combined Notch and IL-23 generate synergistic effects on differentiation of mature ILC subsets from multi-potent as well as uni-potent ILCP.

To understand unique effects of Notch signaling in our system, we compared clonal ILC outcomes on OP9 versus OP9-DLL4 stroma. In the absence of IL-23, DLL4 promoted significantly more ILC2 (mean increase of 27%) (Supplementary Fig. 6c, Supplementary Table 5), matching the increase frequency of total IL-13 producing clones (27%) (Supplementary Fig. 6g, Supplementary Table 5). While overall frequencies of multi-potent clones did not change (Supplementary Fig. 6c, Supplementary Table 5), a modest but significant increase in ILC2/ILC3 clones (mean of 3%) was observed in the presence of DLL4 (Supplementary Fig. 6e, Supplementary Table 5).

We next assessed effects of Notch signaling in the presence of IL-23. Notch signaling resulted in a mean increase of 24% in the frequency of multipotent clones (Supplementary Fig. 6d, Supplementary Table 5), due to increases in multipotent clones containing ILC3 (mean increase of 4, 8 and 6% for Group 1/ILC3, ILC2/ILC3, Group 1/ILC2/ILC3 respectively, overall increase of 20%) (Supplementary Fig. 6f, Supplementary Table 5). Notch signaling in the presence of IL-23 also resulted in an average reduction of 9% in the frequency of EOMES⁺Group1 clones (Supplementary Fig. 6d, Supplementary Table 5). Irrespective of the presence or absence of IL-23, Notch signaling itself reduced the efficiency of cloning by an average of 15% (Supplementary Fig. 6j, Supplementary Table 7), as well as the frequency of the 'non-producers', by 15% in the absence of IL-23 (Supplementary Fig. 6c, Supplementary Table 5) and 19% in the presence of IL-23 (Supplementary Fig. 6d, Supplementary Table 5). Notch signaling therefore appears to selectively permit efficient generation of multi-potent ILCP and additionally promotes ILC2 differentiation. The presence or absence of IL-23 had no effect on ILCP cloning efficiency (Supplementary Fig. 6i, Supplementary Table 7), suggesting that it does not enhance the proliferation of specific biased ILC precursors, but promotes their acquisition of effector functions.

### Human ILCP development and differentiation in IL12RB1 deficient patients

To confirm the important role for IL-23 signaling in ILCP differentiation, we studied *IL12RB1⁻/⁻* patients[28–30,39]. Previous reports have shown that mutations in *IL12RB1* impair Th1 differentiation and IFN-γ production in response to IL-12 and IL-23, resulting in MSMD and increased susceptibility to intracellular pathogens including *Salmonella spp.*, but can also impair Th17 cell development and immunity to fungal pathogens such as *Candida albicans*[26,40]. We first analyzed peripheral blood ILC subsets in patients lacking *IL12RB1* and were able to identify ILCP, ILC2 and NK cells that were present at similar (ILCP, NK cells) or slightly increased levels (ILC2) compared to healthy donors (Fig. 6a–d, Supplementary Fig. 7a, b, Supplementary Table 8). As a proportion of total CD127⁺ ILC, both ILCP and ILC2 were present at frequencies comparable to healthy donors (Fig. 6f, g, Supplementary Table 8). ILCP from *IL12RB1⁻/⁻* patients had a normal phenotype with typical CD25, CD200R1, CD62L, CD161, CD45RA and CD56 expression (Fig. 6h, Supplementary Fig. 7c) and clustered with healthy donor ILCP after UMAP analyses (Fig. 6i).

ILCP cloning from *IL12RB1⁻/⁻* patients on OP9 and OP9-DLL4 stroma with IL-23 gave rise to significantly fewer (average of 10%) EOMES⁺Group 1 ILC clones and significantly more ILC2 clones in both conditions, (average of 21% on OP9, average of 28% on OP9-DLL4) compared to healthy donors (Fig. 7a, b, Supplementary Table 5). Moreover, no Group 1/ILC3 were generated from any *IL12RB1⁻/⁻* patient (Fig. 7c, d), and the cumulative distributions of all clones generated from healthy donors in comparison to *IL12RB1⁻/⁻* patients were significantly different (Fig. 7e, f, Supplementary Table 6; chi-squared test = $p > 0.0001$). ILCP from *IL12RB1* patients were unable to generate significant proportions of IL-17A or IL-22 producing clones even in the presence of Notch signaling (Supplementary Fig. 8a, b, Supplementary Table 5). Finally, ILCP cloning efficiency from *IL12RB1⁻/⁻* patients was similar to that observed using ILCP from healthy donors (Supplementary Fig. 8c, Supplementary Table 7). Taken together, our results suggest that Notch signaling alone does not appear to be sufficient for the generation of ILC3 fate but requires integration of IL-23 signaling to promote multi-lineage ILC differentiation from ILCP.

## Discussion

The unique and cooperative roles for Notch and IL-23 signalling in promoting human ILCP differentiation remain poorly defined and here we provide evidence that the transcription factor RORγT plays a key intermediary role in this process. Previous studies have shown that murine Th17 and ILC3 differentiation both require RORγT[13,26,27], although the precise mechanisms have not been elucidated. RORC controls IL-23R expression[41,42] but mice lacking IL-23 signaling retain gut ILC3 subsets, suggesting alternative regulatory pathways. Pharmacological inhibition of RORC has also produced conflicting results with ablation of Th17 function but little effect on ILC3 differentiation[43]. Our results using RORC-deficient patients and RORγT inhibitors clearly point to an important role for RORC-dependent signals in generating mature functional human ILC3. This included not only IL-17A-producing ILCs but also extended to IL-22-producing ILC3, albeit to a lesser extent. The distinct impact of RORC inhibition for IL-17A versus IL-22 production in human ILC3 may involve alternative regulators of IL-22 expression, such as AHR, which also promotes IL-22 expression in ILC3[44–46]. Moreover, we found an important impact of RORC inhibition on differentiation of multi-potent ILC clones. These results suggest a broader role for RORγT in early human hematopoietic precursors as previously proposed[20,27,47].

Notch signaling plays a key role in T cell development in the thymus and T cell differentiation in secondary lymphoid tissues, where specific Notch ligands act as instructors and/or amplifiers of T cell fate and function[48]. DLL4 has been reported to enhance Th1 differentiation and inhibit Th2 differentiation[49,50], and may also promote Th17 differentiation through RORγT stabilization[51–53]. However, other reports have documented that Notch signaling can simultaneously enhance Th1, Th2 and Th17 differentiation in the absence of polarizing cytokines[54]. Concerning ILC differentiation, Notch signaling has been implicated in promoting ILC2 and ILC3 development in mice, and in human hematopoietic precursors isolated from secondary lymphoid tissues[55–59]. Still, how Notch signaling operates, as either a general enhancer of ILC differentiation or as a promoter of particular ILC subsets remains unclear. Our finding that Notch stimulation up-

 

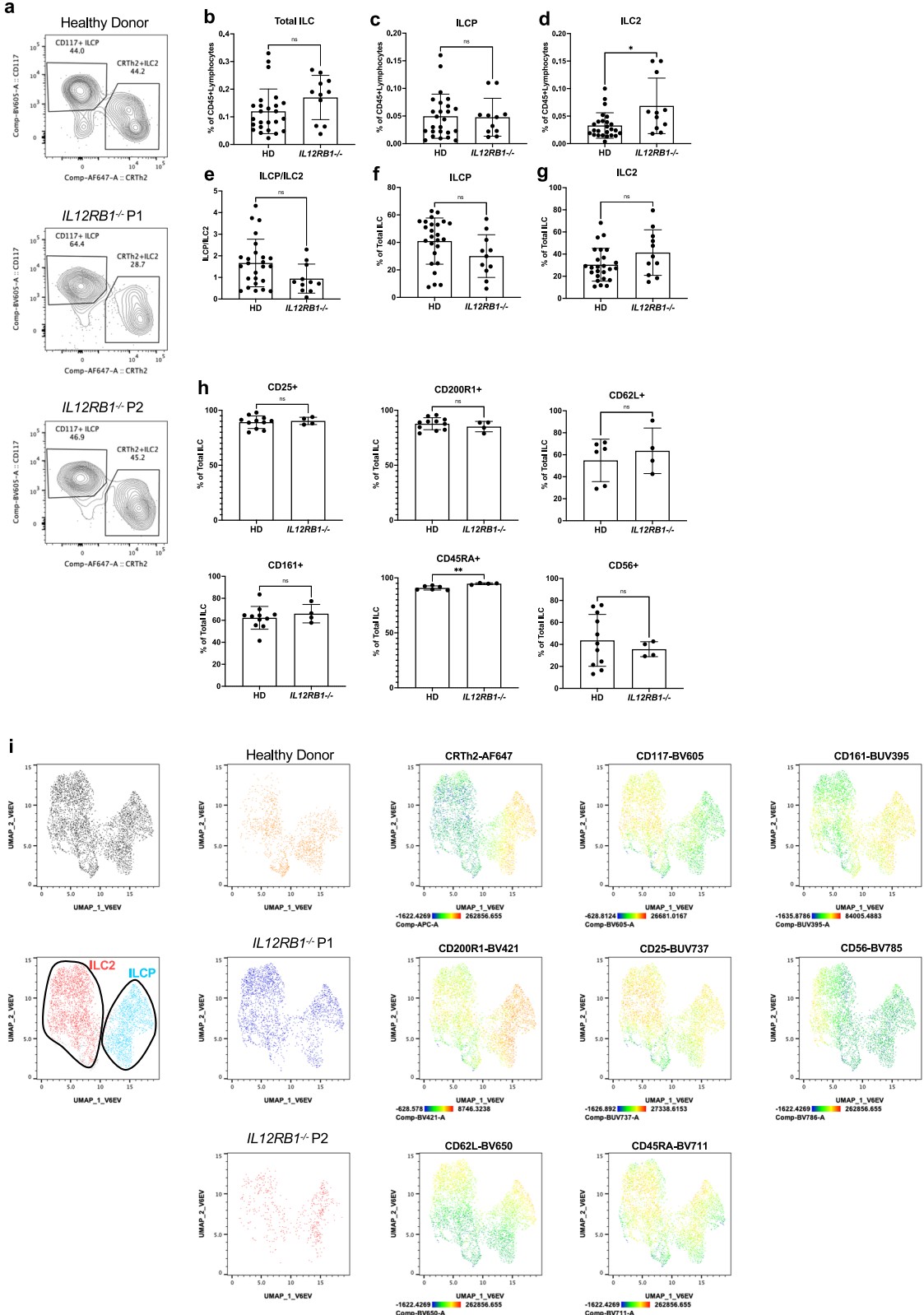

**Fig. 6 | Human ILC differentiation in *IL12RB1*⁻/⁻ patients. a** Flow cytometric analysis of ILCs from a healthy donor and two *IL12RB1*⁻/⁻ patients (gating strategy in Supplementary Fig. 1). Relative frequencies of **b** total ILC (**c**) ILCP (**d**) and ILC2 in CD45+ live lymphocytes. **e** Ratio of ILCP to ILC2. Relative frequencies of **f** ILCP and **g** ILC2 in total ILCs. **h** Expression of markers CD25, CD200R1, CD62L, CD161, CD45RA and CD56 on ILCP of healthy donors and *IL12RB1*⁻/⁻ patients. **b**–**g** Comparisons done using two-tailed Mann–Whitney test. ns = not significant,

*$p \le 0.05$, **$p \le 0.01$, ***$p \le 0.001$, ****$p < 0.001$. **i** Uniform manifold projection (UMAP) analyses on one healthy donor and two *IL12RB1*⁻/⁻ patients. Representative plots of staining can be found in Supplementary Fig. 7 (**b**–**h**). Data are represented as Mean with SD and each point corresponds to an individual donor or patient. **b**–**g** $n = 25$ healthy donors and $n = 11$ *IL12RB1*⁻/⁻ patients, **h** $n = 6$ or 11 healthy donors and $n = 4$ *IL12RB1*⁻/⁻ patients. Details for each test are found in Supplementary Table 8. Source Data are provided as a Source Data file.

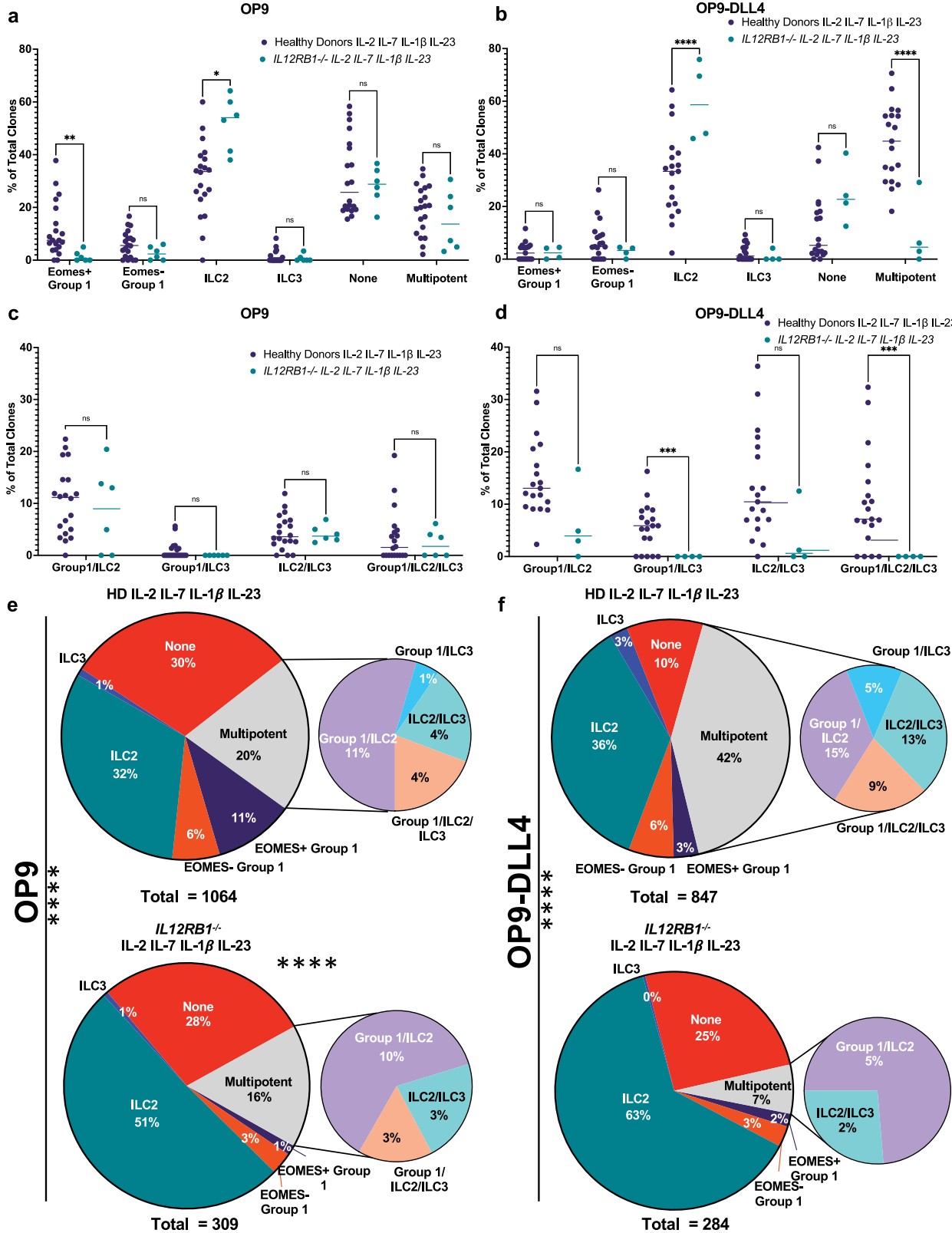

**Fig. 7 | IL12RB1 deficiency results in reduced frequency of multipotent and ILC3 cytokine producing ILC clones.** ILCP from healthy donors (purple circles) or *IL12RB1*−/− patients (green circles) were cloned on OP9 (HD, *n* = 20; *IL12RB1*−/−, *n* = 6) or OP9-DLL4 (HD, *n* = 19; *IL12RB1*−/−, *n* = 4) stroma. Frequencies of unipotent (EOMES + ILC1, EOMES-ILC1, ILC2, ILC3, None) and total multipotent clones per individual healthy donor on **a** OP9 **b** or OP9-DLL4 stroma. Frequencies of types of multipotent clones on **c** OP9 **d** or OP9-DLL4 stroma. Data represented as individual matched values; each point corresponds to an individual donor. Overall distributions of clones grown in the presence or absence of IL-23 on **e** OP9 or **f** OP9-DLL4 stroma. Comparisons performed using (**a**–**d**) Two-way ANOVA with no matching using Šidák's multiple comparisons test, details in Supplementary Table 5 and **e**, **f** chi-square test (observed vs expected; details in Supplementary Table 6). Data are compiled from a minimum of 3 independent experiments. ns = not significant, \**p* ≤ 0.05, \*\**p* ≤ 0.01, \*\*\**p* ≤ 0.001, \*\*\*\**p* < 0.001. Source Data are provided as a Source Data file.

regulated RORC expression in human ILCP offers an explanation for the broad effects of Notch stimulation on human ILC differentiation.

Interestingly, IL-23 signaling enhanced the differentiation of Group 1 and Group 3 ILC from human ILCP, independent of Notch signaling. The impact on IFN-γ expressing Group 1 ILCs is relevant considering that defects in IFN-γ-dependent anti-mycobacterial immunity is observed in patients with mutations in *IL23R*, *IL12RB1* and *RORC*[27,28]. Further, ILCP from *IL12RB1*[−/−] patients primarily gave rise to ILC2 and had clear reductions in the generation of ILC3. While IL-23 affected ILCP differentiation irrespective of the presence of DLL4, Notch signaling clearly enhanced IL-23 mediated effects. That RORγT is required for the action of IL-23 to promote ILC3 differentiation is supported by both its upregulation by Notch signaling and the decline in ILC3 development during RORC inhibition. Effects on ILC3s parallel Th17 differentiation, during which RORγT upregulates *IL23R*; IL-23 signaling and the phosphorylation and activation of STAT3 is subsequently essential for the differentiation and function of Th17 cells, as evidenced by defects in Th17 differentiation and immunity in patients with mutations in *IL23R, IL12RB1* and *STAT3*[26,28,40]. Still, Notch signals exhibit additional effects through stabilization of RORγT expression and IL-23 responsiveness that maintains ILCP multipotency. These data therefore identify a core Notch -> RORC - > IL-23R pathway that regulates human ILC differentiation at multiple stages.

We previously proposed an 'ILC-poiesis' model in which circulating ILCP are poised for further differentiation with ILC maturation relying on tissue signals that will vary depending on the state of stress, infection or inflammation[22,23]. Our data here identify IL-23 as one such environmental factor that can modify ILC fate from ILCP by promoting multi-potent ILCP differentiation. We did observe that a proportion of ILCP in all conditions gave rise exclusively to a single fate, primarily ILC2, which may indicate that a proportion of ILCP are unipotent ILC2P[32]. Alternatively, ILC2 may represent a 'default' ILC maturation pathway when certain environmental signals are lacking. Such 'homeostatic' ILC2 may be important in the maintenance of tissues under steady-state conditions and during repair after inflammatory insults.

Multi-potent ILCP may have enhanced capacity to adapt to the tissue environment into which they are recruited. The tissue distribution of Notch ligands (and DLL4 in particular) may represent an important factor in generating tissue-specific repertoires of ILCs. ILC3 are not found in the circulation, and are also rare in many non-mucosal tissues[22,60]. Tonsils are enriched in Notch ligands, and are one of the tissues in which ILC3 have been most studied in humans, in part due to their relative abundance, DLL4 is also highly expressed in the human small intestine and other tissues of the digestive system[58,61]. This is notably also one of the tissues in which ILC3 are most abundant, both in human and in mice[13,60,62], and where they have been demonstrated to act in the maintenance of gut homeostasis and responses to pathogenic bacteria. A synergistic effect for IL-23 and Notch signaling may be particularly relevant in the intestine. IL-23 is expressed by dendritic cells and antigen presenting cells in the intestine and is important in the maintenance and homeostasis of epithelial barrier function[3,63]. Thus, Notch and IL-23 signaling may act as tissue specific signals which direct the development of ILC upon entry into the tissue and may thereby promote the generation of ILC3.

The ability of ILCP to flexibly generate ILC subsets dictated by external signals may also have important consequences in the early immune response to infection. Notably, patients with LOF mutations in *IL12RB1* or *RORC* both are susceptible to chronic mucocutaneous candidiasis (CMC) resulting from infection with the common opportunistic fungal pathogen *Candida albicans*[40,64]. Th17 generation and function is impacted in these patients as well; this defect in T cell differentiation is likely a key factor in the development of fungal disease in these patients[26]. ILC3 have also been demonstrated to exhibit anti-fungal immunity through the expression of IL-17A, and due to

their localization in mucosal barrier tissues where pathogen exposure typically occurs, may play important roles in the initial immune response[65]. While the relative distributions of ILCs in tissues from both *RORC* and *IL12RB1* patients is not known, the observed defects in their generation of Group 3 ILCs from ILCP may contribute to their susceptibility to fungal infections. It is also interesting to note that while a significant proportion of *IL12RB1* patients develop candidiasis, it is relatively more common, severe and chronic in patients with *RORC* deficiencies[22,26–28]; this may correlate with the greater impacts on both ILCP and ILC3 generation seen in RORγT-deficient patients compared to IL12RB1-deficient patients. While the differences in ILC3s were the most striking in our investigation of the *RORC*/IL-23 axis, we also observed clear impacts on the generation of IFN-γ-producing Group 1 ILCs. This is in line with research demonstrating that mutations in *RORC* as well as both the IL-12 and IL-23 signaling axes are also strongly associated with MSMD as a result of impaired IFNγ-dependent immunity to mycobacteria[27–29]. The roles of ILCs and of ILCP differentiation in the context of mycobacterial infection are not well defined, however impairment of the generation or function of IFN-γ-producing Group 1 ILCs may further permit mycobacterial proliferation early in MSMD.

In summary, we have demonstrated key roles for Notch DLL4 signals, RORC and IL-23 in promoting multi-potential ILC fate, and for the generation of IL-17A and IL-22-producing ILC3. While future work will be required to define the precise molecular mechanisms underlying these processes, our results demonstrate that manipulation of these pathways can profoundly modify ILC differentiation, providing a potential path to harness the therapeutic potential of these novel effector cells in the clinic.

## Methods

### Human samples and cell isolation
All studies required ethics approval from the institutional ethics committees. Patient recruitment was done under the ID-RCB 2010-A00650-39 and 2010-A00636-33 delivered by EC IDF-II, France. Healthy donors were recruited by Etablissement Français du Sang (EFS, Paris, France) and samples given to the Institut Pasteur (Paris, France) under the agreement N°18/EFS/041. Healthy donors were provided without preselection and ranged in age from 19–61 with a 60–40 ratio of males to females. Essential information regarding patients with autosomal recessive complete IL-12Rβ1 and RORγT deficiency are included in Supplementary Table 3 and has been previously described[27,39,64,66–68]. Informed consent was obtained from all patients, including protocols approved by the institutional review boards. A Ficoll-Paque gradient (GE Healthcare) was used for PBMC isolation. Cells from patients and healthy donors were studied after cryopreservation unless fresh samples were available.

### Isolation and phenotyping of ILCP
For bulk sorting, PBMCs were first depleted of T cells, B cells, and residual monocytes, eosinophils, pDCs, and erythrocytes by incubation with biotinylated antibodies (anti-CD3, anti-CD14, anti-CD19, anti-CD123, anti-CD235a) for 20–30 min at room temperature, followed by incubation with MojoSort anti-biotin magnetic nanobeads (Stemcell) according to the manufacturer's instructions. Samples used for single cell cloning were thawed directly from cryopreserved PBMCs. Samples were stained for 15 min at room temperature in 2%FCS-PBS for lineage markers (anti-CD3, anti-CD4, anti-CD5, anti-CD14, anti-CD19, anti-TCRαβ, anti-TCRγδ), anti-CD45, anti-CD7, anti-CD127, anti-CD16, anti-CD94, anti-NKG2A, anti-CD117, anti-CRTh2, as well as other markers used for surface phenotyping. A complete list of antibodies can be found in Supplementary Table 9. Cells were sorted and samples acquired simultaneously using a FACS Aria II (BD) using BD FACS DIVA version 6 (BD). or FACS Aria Fusion (BD) using BD FACS DIVA version 8 (BD). Bulks were sorted to >99% purity and single cell cloning was performed using single cell purity and index sorting.

## Cell culture

All experiments performed in vitro were performed using Yssel's media: IMDM + Glutamax (ThermoFisher scientific), BSA 0.25% (w/v) (Sigma), 2-aminoethanol (1.8 µg/L) (Sigma), Apo-transferrin (40 µg/L) (Sigma), Insulin (5 µg/L) (Sigma), supplemented with 2% human AB serum (EFS), in wells pre-seeded the night before with ~2000 OP9 or OP9-DLL4 stromal cells[22]. The cytokines IL-2, IL-7 (Miltenyi Biotec), IL-1β and IL-23 (Peprotech) were added in the combinations indicated at 50 ng/ml for bulk culture and 10 ng/ml for single cell cloning. For bulk culture, 1000–3000 FACS sorted cells were plated onto stromal cells. For cloning, cells were directly sorted into the 96-well plates using index sorting. Media was changed every 2–3 days for bulk cultures and every 4–5 days for single cell cloning. Bulk cultures were analyzed after 5–7 days and cloning experiments after 13 days.

## RORC inhibition

The specific RORC inhibitor SR2211[36] was purchased commercially (Tocris/Biotechne) and resuspended in DMSO at a stock of 100 mM. SR2211 was added at a final concentration of 10 µM in both bulk and single cell cultures; controls were treated with the equivalent concentration of DMSO (0.01%). SR2211 toxicity was tested on growing ILC cultures using a range of 5–100 µM; no effect was seen at 10 µM which was the dose previously reported[36]. SR2211 and control DMSO were replenished every 2–3 days for bulk cultures and every 4–5 days for single cell cloning.

## Intracellular staining and functional analysis

Analysis of ILC cytokine production was performed by intracellular staining after 3 h of pharmacological stimulation with PMA (10 ng/ml; Sigma), ionomycin (1 ug/ml; Sigma) and Golgi Plug (Brefeldin A; BD)[22]. Samples taken for transcription factor and phenotype analysis were taken directly for staining. Cells were stained for extracellular markers with Fc block and viability dye efluor 506 (Thermofisher Scientific) for 30 min at 4 °C in Brilliant Stain Buffer (BD). Fixation and permeabilization was performed using the FoxP3/Transcription Factor staining kit (Thermofisher Scientific) as per the manufacturer's instructions. Cells were then stained for 30 min at room temperature for intracellular and intranuclear markers as indicated in Supplementary Table 9. Samples were acquired on an LSR Fortessa (BD) using BD FACS DIVA version 8 (BD) and analyzed using FlowJo (BD).

## Antibodies

All antibodies used in this study are pre-conjugated and commercially available. A complete list of antibodies can be found in Supplementary Table 9.

## Statistical and data analysis

Statistical analysis was performed using GraphPad Prism version 9.2.0 for Mac OS X, GraphPad Software, San Diego, California USA, www.graphpad.com. Unless otherwise detailed, all tests are two tailed; details of all multiple comparison tests including degrees of freedom, number of individuals per condition and either F or t statistics are included in Supplementary Tables 1, 2, 4–8. The details and assumptions of all tests are described in the relevant figure legends or in the referenced supplementary tables. The axes of all scatterplots were designed to include all data points. Analysis of flow cytometry data was performed using FlowJo (BD) software version 10.8.0 for MAC OS X https://www.flowjo.com/. Uniform manifold projection (UMAP) analyses[35] were performed using the FlowJo (BD) UMAP plugin version 3.1. Data are represented as mean with SD unless specified. Sample size and number of experiments denoted in figure legends.

## Reporting summary

Further information on research design is available in the Nature Research Reporting Summary linked to this article.

## Data availability

All data generated or analyzed during this study are available as Supplementary Data. Source data are provided with this paper.

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

## Acknowledgements

We are indebted to the patients and their families for their essential collaboration. We thank members on Innate Immunity Unit for the helpful discussions, and the Center for Translational Science (CRT)-Cytometry and Biomarkers Unit of Technology and Service (CB UTechS) at Institut Pasteur for their technical support. Funding was provided by Institut Pasteur, the Institut National de la Santé et de la Recherche Médicale (Inserm), the European Research Council (ERC) under the European Union's Horizon 2020 research and innovation program (695467 – ILC_REACTIVITY), the National Institute of Allergy and Infectious Diseases, NIH (R01AI095983) and the French National Research Agency (ANR) GENMSMD (ANR-16-CE17-0005-01) to J.P.D. C.A.C. is enrolled in the Pasteur-Paris University (PPU) International PhD Program that receives funding from the European Union's Horizon 2020 research and innovation programme under the Marie Sklodowska-Curie grant agreement #665807, from the Labex Revive (10-LABX-0073), Institut Pasteur and the Fondation ARC. A.T. received funding from the European Union's Horizon 2020 research and innovation programme under the Marie Skłodowska-Curie grant agreement #765104.

## Author contributions

Conceptualization, C.A.C. and J.P.D.; Methodology, C.A.C., A.T., S.M., J.-M.D., L.S., R.Y.; Formal Analysis, C.A.C.; Investigation, C.A.C., A.T., S.M., J-M.D., L.S., R.Y.; Resources, J.B., A.P., J.-L.C.; Writing, C.A.C., J.P.D.; Funding Acquisition, J.P.D.; Supervision, J.P.D.

## Competing interests

The authors declare no competing interests.
