## [Peer Review File · Nature Communications]

Notch, RORC and IL-23 signals cooperate to promote multi-lineage human innate lymphoid cell differentiationREVIEWERS' COMMENTS

Reviewer #1 (Remarks to the Author):

This is an important study that examines the differentiation of human ILC. Using a number of elegantly designed experiments and rare samples, the authors describe cooperative roles for the Notch and IL-23 signaling pathways in regulating human ILC differentiation. The significance and novelty are high. There are a few concerns regarding the rigor of the studies.

- (1) Essential details of the healthy controls, such as sex and age, should be provided.
- (2) More details for the patients, such as sex, disease conditions and medication, should be provided.
- (3) Results in Figure 2 are based on data from 2 RORC deficient patients. The samples are precious and the results are informative. However, because the sample size is too small and is not a statistically valid sample size, it is not appropriate to calculate P values for these data.
- (4) In Figure 4, only 1 dose (10uM) was used. Did the authors do dose searching? How did the authors verify inhibition efficiency?
- (5) In Figures 6 and 7, does deficiency of IL-12R affect Rorc expression? Could the authors possibly use age and sex matched healthy control with equal sample size?

Reviewer #2 (Remarks to the Author):

In this manuscript, Croft and colleagues report the effect of Notch signaling and IL-23 on human ILC differentiation. The strengths of this study are the sole use of human ILC precursors (ILCP) in the experiments and the inclusion of ILCP from human knockouts for RORC and IL-23RB1. This is a major benefit because of this is one of the few groups in the world that have access to such patients. The logic behind the experiments is well justified, and given the expertise of this group, it is not surprising that the data is clear and convincing. The experiments in which ILC are cultured in OP9 and OP9-DLL4 give a clear picture of the role of Notch signaling in ILCP differentiation. Notch was first reported to be important in ILC3 development by Song and colleagues (J Exp Med. 212(11):1869-82, 2015) and IL-23 has been reported to enhance ILC3 development by several groups. Therefore, it is not a stretch to anticipate that the combination of Notch signaling with IL-23 would enhance RORC/RORγT expression in developing ILC and the generation of ILC3 from peripheral blood ILCP. Likewise, one would not anticipate IL-23 to have an effect in promoting ILC3 from ILCP from patients with deletion of IL-12RB1, since this receptor subunit is a component of the IL-23 receptor.

Major points

Lines 162 and 185 – Were the RORC^{-/-} patients the same for both of these experiments? If so, the authors should indicate this.

Figure 2 – It is difficult to draw firm conclusions when performing statistics on a group that contains only 2 data points.

Minor points

Line 173 – Delete the word “into” so that the sentence reads “ILC clones were then classified based on cytokine expression.”

Reviewer #3 (Remarks to the Author):

Croft et al have used bulk and single cell cloning assays to investigate the influence of Notch, IL-23 and RORgT molecules on the development of ILC subsets from multilineage human innate lymphoid cells, which they first identified and reported in a seminal publication in Cell in 2017. The results herein represent a huge effort. The data is certainly interesting, but quite descriptive and in general only an incremental advance on current understanding of ILC development. The role of Notch in human ILC development is well known and it is not clear that the current ms adds significantly to this area. Similarly, the roles of RORgT are also well described. Finally, the impact of cytokines including IL-23 have been intensively studied. Thus, many of the results are confirmatory of previous published work.

Additional comments

1. The comparison of 18 samples with 2 samples is not ideal.
2. Line 145 claims synergistic effect but this is not the case for RORgT.
3. Supp Fig 5 h is on the figure image but not included in the figure legend or text. The results in Supp Fig 5 f and h appear to indicate an increase in IL-22.
4. Line 245. Fig 5e comes before Fig 5d and Fig 5f is not mentioned in text.
5. Figure text sizes are too small in main figures and especially supplementary.

NCOMMS-21-50471

Notch, RORC and IL-23 signals cooperate to promote multi-lineage human innate lymphoid cell differentiation

Reviewer #1:

This is an important study that examines the differentiation of human ILC. Using a number of elegantly designed experiments and rare samples, the authors describe cooperative roles for the Notch and IL-23 signaling pathways in regulating human ILC differentiation. The significance and novelty are high. There are a few concerns regarding the rigor of the studies.

We thank the reviewer for their supportive comments on our work.

(1) Essential details of the healthy controls, such as sex and age, should be provided.

We have added details on healthy control age/sex to the revised manuscript (Materials and methods).

(2) More details for the patients, such as sex, disease conditions and medication, should be provided.

We have included additional details for patients, including sex, disease conditions, and medications as requested (revised Supplemental Table 1).

(3) Results in Figure 2 are based on data from 2 RORC deficient patients. The samples are precious and the results are informative. However, because the sample size is too small and is not a statistically valid sample size, it is not appropriate to calculate P values for these data.

We agree that the results in Figure 2 using RORC deficient patients is limited and thank the reviewer for stating nevertheless that the results are informative. We have removed the statistical analysis.

(4) In Figure 4, only 1 dose (10uM) was used. Did the authors do dose searching? How did the authors verify inhibition efficiency?

We tested a range of doses (5 to 100 uM) of the RORgT inhibitor in preliminary studies using growing human ILC clones and assessing toxicity. We found no toxicity at 10 uM concentration and as this corresponded to the previously reports showing inhibition of RORgT, we used this concentration in our experiments.

(5) In Figures 6 and 7, does deficiency of IL-12R affect Rorc expression? Could the authors possibly use age and sex matched healthy control with equal sample size?

Several patients were quite young at sampling and age-matched controls were not available. We did not observe any obvious sex- or age-related bias in our results. We did not assess RORC transcript expression in these samples as we prioritized the ILCP phenotypic and cloning analyses.

Reviewer #2

In this manuscript, Croft and colleagues report the effect of Notch signaling and IL-23 on human ILC differentiation. The strengths of this study are the sole use of human ILC precursors (ILCP) in the experiments and the inclusion of ILCP from human knockouts for RORC and IL-23RB1. This is a major benefit because of this is one of the few groups in the world that have access to such patients. The logic behind the experiments is well justified, and given the expertise of this group, it is not surprising that the data is clear and convincing. The experiments in which ILC are cultured in OP9 and OP9-DLL4 give a clear picture of the role of Notch signaling in ILCP differentiation. Notch was first reported to be important in ILC3 development by Song and colleagues (J Exp Med. 212(11):1869-82, 2015) and IL-23 has been reported to enhance ILC3 development by several groups. Therefore, it is not a stretch to anticipate that the combination of Notch signaling with IL-23 would enhance RORC/ROR γ T expression in developing ILC and the generation of ILC3 from peripheral blood ILCP. Likewise, one would not anticipate IL-23 to have an effect in promoting ILC3 from ILCP from patients with deletion of IL-12RB1, since this receptor subunit is a component of the IL-23 receptor.

We thank the reviewer for their positive comments concerning our approach and presented results. While we agree that some results may have been anticipated, it is important to test such hypotheses with rigorous experimentation (which we have tried to do) in order to make validated conclusions.

Major points

Lines 162 and 185 – Were the RORC $^{-/-}$ patients the same for both of these experiments? If so, the authors should indicate this.

We now indicate that the same two RORC $^{-/-}$ patients were used for both ILCP phenotyping and cloning analyses.

Figure 2 – It is difficult to draw firm conclusions when performing statistics on a group that contains only 2 data points.

A similar point was raised by Reviewer #1, comment 3. We have removed the statistical analysis.

Minor points

Line 173 – Delete the word “into” so that the sentence reads “ILC clones were then classified based on cytokine expression.”

We have corrected the sentence as requested.

Reviewer #3:

Croft et al have used bulk and single cell cloning assays to investigate the influence of Notch, IL-23 and RORgT molecules on the development of ILC subsets from multilineage human innate lymphoid cells, which they first identified and reported in a seminal publication in Cell in 2017. The results herein represent a huge effort. The data is certainly interesting, but quite descriptive and in general only an incremental advance on current understanding of ILC development. The role of Notch in human ILC development is well known and it is not clear that the current ms adds significantly to this area. Similarly, the roles of RORgT are also well described. Finally, the impact of cytokines including IL-23 have been intensively studied. Thus, many of the results are confirmatory of previous published work.

We thank the reviewer for their comments concerning the effort involved in our study and their frank comments concerning the significance. We would only reply that our approach using pharmacological inhibition and analyses of ILCPs from patients with rare genetic mutations have never been previously performed and provide coherent results showing that these approaches can be applied by others in the field who may not have access to the rare patient samples.

Additional comments

1. The comparison of 18 samples with 2 samples is not ideal.

A similar point was raised by Reviewers #1 and #2. We have removed the statistical analysis.

2. Line 145 claims synergistic effect but this is not the case for RORgT.

We have modified the sentence to read: "Taken together, these results suggest an effect of Notch signaling via DLL4 on RORC/RORgT expression in developing ILC and a synergistic role for IL-23 and Notch signaling for the generation of Group ILC1 and ILC3 from peripheral blood ILCP."

3. Supp Fig 5 h is on the figure image but not included in the figure legend or text. The results in Supp Fig 5 f and h appear to indicate an increase in IL-22.

We now refer to this data (Supp Fig 5h) in the revised manuscript (page 7, line 209). The frequency of IL-22+ cells was decreased in the presence of SR2211 (9.9% versus 2.2%).

4. Line 245. Fig 5e comes before Fig 5d and Fig 5f is not mentioned in text.

The call-out sequence for Figure 5 panels has been corrected.

5. Figure text sizes are too small in main figures and especially supplementary.

We have increased the font sizes in the figures to help make them easier to read.

REVIEWERS' COMMENTS

Reviewer #1 (Remarks to the Author):

The authors have addressed all the comments.

Reviewer #2 (Remarks to the Author):

The authors have sufficiently addressed the critiques that I submitted.

Reviewer #3 (Remarks to the Author):

No further comments

NCOMMS-21-50471B

Notch, RORC and IL-23 signals cooperate to promote multi-lineage human innate lymphoid cell differentiation

Reviewer #1 (Remarks to the Author):

The authors have addressed all the comments.

Reviewer #2 (Remarks to the Author):

The authors have sufficiently addressed the critiques that I submitted.

Reviewer #3 (Remarks to the Author):

No further comments

We thank the reviewers for their time and effort in evaluating our manuscript, for their helpful suggestions and for their supportive comments on our work.